# WGCNA Identifies Translational and Proteasome-Ubiquitin Dysfunction in Rett Syndrome

**DOI:** 10.3390/ijms22189954

**Published:** 2021-09-15

**Authors:** Florencia Haase, Brian S. Gloss, Patrick P. L. Tam, Wendy A. Gold

**Affiliations:** 1Faculty of Medicine and Health, School of Medical Science, The University of Sydney, Camperdown, NSW 2050, Australia; florencia.haase@sydney.edu.au (F.H.); ptam@cmri.org.au (P.P.L.T.); 2Kids Neuroscience Centre, Kids Research, Children’s Hospital at Westmead, Westmead, NSW 2145, Australia; 3Molecular Neurobiology Research Laboratory, Kids Research, Children’s Hospital at Westmead, Westmead, NSW 2145, Australia; 4Westmead Research Hub, Westmead Institute for Medical Research, Westmead, NSW 2145, Australia; brian.gloss@sydney.edu.au; 5Embryology Research Unit, Children’s Medical Research Institute, The University of Sydney, Sydney, NSW 2145, Australia

**Keywords:** RTT, iPSCs, WGCNA

## Abstract

Rett Syndrome (RTT) is an X linked neurodevelopmental disorder caused by mutations in the methyl-CpG-binding protein 2 (*MECP2*) gene, resulting in severe cognitive and physical disabilities. Despite an apparent normal prenatal and postnatal development period, symptoms usually present around 6 to 18 months of age. Little is known about the consequences of MeCP2 deficiency at a molecular and cellular level before the onset of symptoms in neural cells, and subtle changes at this highly sensitive developmental stage may begin earlier than symptomatic manifestation. Recent transcriptomic studies of patient induced pluripotent stem cells (iPSC)-differentiated neurons and brain organoids harbouring pathogenic mutations in MECP2, have unravelled new insights into the cellular and molecular changes caused by these mutations. Here we interrogated transcriptomic modifications in RTT patients using publicly available RNA-sequencing datasets of patient iPSCs harbouring pathogenic mutations and healthy control iPSCs by Weighted Gene Correlation Network Analysis (WGCNA). Preservation analysis identified core gene pathways involved in translation, ribosomal function, and ubiquitination perturbed in some MECP2 mutant iPSC lines. Furthermore, differential gene expression of the parental fibroblasts and iPSC-derived neurons revealed alterations in genes in the ubiquitination pathway and neurotransmission in fibroblasts and differentiated neurons respectively. These findings might suggest that global translational dysregulation and proteasome ubiquitin function in Rett syndrome begins in progenitor cells prior to lineage commitment and differentiation into neural cells.

## 1. Introduction

Rett syndrome (RTT) is a severe X-linked neurodevelopmental disorder characterised by loss of fine and gross motor skills, abnormal social behaviour, growth retardation, seizures, and breathing dysregulation [1]. RTT predominantly affects females, with a worldwide prevalence of 1 in 10,000 girls [2]. The majority of RTT cases are caused by *de novo* mutations in the X-linked methyl-CpG-binding protein 2 gene (MECP2) encoding the MeCP2 protein, which plays a critical role specifically in the maturation of the central nervous system (CNS) and synaptic function [3,4]. Mutations that result in the loss of MeCP2 function cause a range of molecular, cellular, and anatomical abnormalities that perpetuate at the symptomatic phase [5].

RTT is associated with more than 500 pathogenic mutations, with the majority being located in three key domains: the methyl-binding domain (MBD), transcriptional-repression domain (TRD), and nuclear localisation signal (NLS) domain [6]. MECP2 plays a critical role in transcriptional regulation, and it is key for normal brain development. While genotype-phenotype correlations are limited, it has been noted that mutations located upstream in the N-terminal and including p.R270* are associated with more severe symptoms when compared with those of p.R294* or C-terminal–truncating mutations [7]. Despite the availability of pre-clinical models of RTT, these disease models do not fully recapitulate all aspects of the human pathology. The limitations in their reliability for understanding disease physiology and the predictive value for clinical outcomes poses a significant hurdle to devising effective therapeutics [7,8]. So far, there are no effective pharmaceutical neuromodulators that alter the course of disease in RTT individuals.

Induced Pluripotent stem cells (iPSCs) reprogrammed from patient somatic cells that can be differentiated into neurons, provide a valuable complementary disease model to existing models. RTT iPSC derived neurons recapitulate the RTT phenotype, such as smaller soma size, reduced dendritic branching and axonal arborisation, and unbalanced excitatory versus inhibitory activity. While many iPSC studies using RTT patient-derived cells have demonstrated their utility as a valuable model, few studies have explored the molecular and cellular attributes and the RTT pathophysiology of the iPSCs at the stem cell state.

Transcriptional and global translational dysregulation as well as proteomic perturbations and reduced global protein synthesis in mouse models and humans is well documented in RTT [9,10,11,12,13,14]. Recent evidence has unravelled reduced global translation in RTT iPSC-derived neurons, including reduced ribosome engagement, compromised mTOR signalling, and altered ubiquitination via NEDD4L E3-ubiquitin ligase, demonstrating a globally disturbed translatome during neurodevelopment [15]. In this study, we used a gene interaction network approach to survey gene expression changes in published transcriptomic datasets of RTT patient fibroblasts, iPSCs, and iPSC-differentiated neurons (GSE21037 [16] GSE51607 [17] and GSE107399 [18]). The three studies analysed different patient cell lines, where the mutations spanned exons 3 and 4 of the MECP2 gene. MT [17] analysed a c.1155del33 mutation located in the C-terminus; OH [18] analysed two mutations; c.705delG and c.1461A>G located in the transcriptional repressor domain (TRD) and the C-terminus respectively; and lastly, TK analysed four patient mutations; c. 473C>T, c.1461A>G, c.916C>T and p.E235fs located in the methyl binding domain (MBD), C-terminus respectively, and the last two in the TRD respectively (Table 1, Appendix A). Weighted Gene Co-expression Network Analysis (WGCNA), designed to identify key biological processes in complex expression data [19] was used to detect correlation patterns of key gene modules and pathways underpinning iPSC biology dysregulated in RTT. In addition, preservation analysis of the MT dataset [17] was performed to determine if any of the modules were conserved across the other two datasets (OH [18] and TK [17]). The lavenderblush module, that was highly preserved across the three studies, identified genes involved in translation and ribosomal function, including many NEDD4-family ubiquitin ligases. These findings point to global dysregulation of translation in the RTT iPSCs prior to commitment and differentiation to neural cell lineage, which reflect the transcriptional disruption of key cellular pathways in undifferentiated neural cells in RTT patients.

## 2. Results

### 2.1. Dataset Selection and Gene Expression Analysis

Publicly available genome-wide transcriptomic datasets of RTT fibroblasts, iPSCs, and iPSC-derived neurons were retrieved from the NCBI Gene Expression Omnibus database. These included: GSE21037 [16] (MT), GSE51607 [17] (TK) and GSE107399 [18] (OH). These three studies used different sequencing platforms, where the MT analysis was performed on the Affymetrix Human Gene 1.0 ST Array, TK on the Illumina Genome Analyzer II, and OH on the Illumina HiSeq 2000 (Figure 1). There were also significant study design differences in each dataset: for the undifferentiated iPSCs, the GSE51607 (TK) and GSE107399 (OH) studies included four different diseased samples (from three and two different RTT patients, respectively) harbouring mutations in the C-terminal and TRD of MeCP2, and three isogenic controls. The GSE21037 (MT) study included one patient sample comprising two MECP2-mutant cell clones (Cln 15 and 18) and three replicates for each clone, as well as three age-sex matched healthy controls. Subsequent differential expression analyses were done using the fibroblast profiles from the MT study, and the iPSC-neuron data from the OH study (Table 1; Appendix A).

The three datasets encompassed patient samples harbouring pathogenic mutations in exon 4 of MECP2, located on the MBD, TRD and C-terminal key protein domains (Appendix A). The position of each mutation and the clinical phenotypes of each patient cell line are shown in Figure 2.

### 2.2. Principle Component Analysis and Sample Clustering

We firstly evaluated global gene expression patterns using hierarchical clustering to rule out outlier samples that would impact the downstream WGCNA analysis (Appendix A). We then use Principal Components Analysis (PCA) to interrogate the expression profile relationships (Figure 3). We observed that the MECP2 mutant cell lines separated clearly from controls in the MT dataset whereas OH and TK datasets exhibited more heterogeneity, with some mutant samples forming distinct clusters. Owing to this apparent heterogeneity, we employed WGCNA, a strategy designed to extract coherent gene expression programs from complex data, to elucidate modules of gene expression in iPSC transcriptomes that could potentially be altered in MECP2 mutant cell lines.

### 2.3. iPSCs Generate Modules of Dysregulated Genes

WGCNA was applied to the gene expression data of the MT, OH and TK iPSC datasets to examine gene groups (gene modules) whose expression profiles were highly correlated across samples. The gene modules contained subsets of genes with similar biochemical and functional properties, thereby enabling the discovery of common dysregulated gene modules and convergent molecular pathways in patients with different mutations. We first constructed co-expression networks on the individual iPSC datasets comparing RTT and controls. Gene expression profiles were classified into modules for each dataset, with the expression levels of each module summarised according to the correlation coefficient, either positive or negative, at significance level of *p* < 0.05. Correlations between mutant and control cell lines in the three datasets are shown, with green for negative correlation (darkest green being most negatively correlated) and red for positive correlation (darkest red being most positively correlated) (Figure 4). Seven significantly expressed gene modules were identified in the MT dataset and one in the TK dataset, whereas the OH dataset did not reveal any significant modules, reflecting the study heterogeneity. Of the seven modules identified in the MT dataset, one module was positively correlated [16], consistent with an overall upregulation of gene expression. Six modules were negatively correlated (green), indicating that the predominant impact of the MECP2 mutation in the iPSCs results in gene repression.

### 2.4. Altered Gene Expression Networks Are Enriched for Specific Biological Processes

To determine which cellular pathways were implicated in the significantly correlated modules (MT: black, floralwhite, mediumorchid, yellowgreen, plum4, mistyrose, and lavenderblush; TK: orange), we tested the modules for pathway enrichment using the Kyoto Encyclopedia of Genes and Genomes (KEGG) [20] pathway analysis (Appendix A). Three out of the seven gene modules from the MT dataset were enriched in at least one KEGG pathway (adj. *p* value < 0.05). The significant module (orange) identified in the TK dataset was not enriched for any cellular pathways. The two enriched modules in the MT dataset, lavenderblush (1824 genes, second largest module) and floralwhite (449 genes, tenth largest module) both were negatively correlated and were selected for further analysis for their likely impact on cellular functions.

Given that the MT dataset was the only dataset to produce significant modules that were also enriched in at least one KEGG pathway, we performed a preservation analysis based on the MT dataset to identify whether genes in the respective module were also preserved in the other two groups. Two enriched modules: lavenderblush and floralwhite were mapped against the TK and OH datasets. A preservation score below 2.5 is considered poor preservation, whereas above 10 is very good preservation, with the in-between being moderate (Figure 5A) [19]. While most of the MT dataset modules were not preserved across the three studies, the lavenderblush module, consisting of 1824 co-ordinately expressed genes, reached a preservation score of 10.5 when compared with the respective modules in the TK dataset (MT vs. TK) and a preservation score of 22 when compared with the OH dataset (MT vs. OH) (Figure 5B) indicating that the lavenderblush module represents a gene network co-expressed in all three studies but not necessarily that the module is altered in the trait. 

To evaluate whether the lavenderblush genes would provide insight into other MECP2-mutants tested, we investigated the clustering of the samples for each dataset according to the expression pattern of genes in lavenderblush by Principal Component Analysis (PCA). RTT and control samples of the MT dataset remained segregated (Figure 5B). In contrast, RTT and controls samples in the OH and TK datasets did not show a clear segregation (Figure 5B), demonstrating that, while the module was preserved in the datasets, the gene expression of the module could not be used to stratify other MECP2 mutations.

### 2.5. Key Cellular Pathways Involved in Protein Translation Are Dysregulated in Rett iPSCs

Interrogation of the dysregulated pathways in the preserved lavenderblush module revealed enrichment of several pathways related to protein translation and degradation. The most significant pathways identified included ribosome biogenesis in eukaryotes (#03008; 22 genes, *p* = 3.00 × 10^−6^), protein processing in the endoplasmic reticulum (#04141; 31 genes, *p* = 0.0003), aminoacyl t-RNA biosynthesis (#00970; 15 genes, *p* = 1.48 × 10^−5^) and the proteasome (#03050; 12 genes, *p* = 0.0003) (Table 2, Figure 6A). The protein-protein interaction networks among the genes in these pathways were then plotted using String (default settings; medium interaction score (0.400), FDR stringency of 5%) [21] (Figure 6B) which showed that these pathways represent distinct but interconnected processes. Together, these data demonstrate that the lavenderblush gene network is comprised of multiple key biosynthetic pathways that are preserved in multiple iPSCs experiments. The dysregulation of this module observed in the MT dataset, whilst not a global feature of all MECP2 mutations tested in this study and based on data from a patient harbouring a c.1155del33 mutation, may suggests a role of these pathways in undifferentiated cells of RTT. To unravel whether these alterations could be linked to the more symptomatic molecular phenotype of RTT, we then investigated their expression changes in parental fibroblast cells and iPSC-derived neuron expression profile.

### 2.6. Ubiquitin Genes Are Dysregulated in Fibroblasts

To determine whether the dysregulated pathways observed in the RTT iPSCs were shared by the parental fibroblast cell lines, we examined the gene expression profile of the preserved dysregulated lavenderblush module in the iPSC cell lines (Figure 7A) and compared them with the differentially expressed genes in the fibroblast cell lines of the MT dataset. When the gene expression of the RTT parental fibroblast cell lines was compared to their controls, we observed 17 genes out of the 1726 in the lavenderblush module to be upregulated (Figure 7B) and 35 downregulated (Figure 7C). After performing pathway enrichment, four genes were implicated in the ubiquitination pathway: NEDD4L, ARRDC4, HCN1, KLHL13, (Table 3) and three genes in Pyridoxal Phosphate pathway: MOCOS, CTH, PSAT1 (Figure 7C). Notably, NEDD4L gene expression perturbations represent a unifying molecular feature of patient cells harbouring a mutation at c.1155del33 and therefore may be implicated in the pathogenesis of RTT.

### 2.7. Dysregulated Genes in iPSC-Induced Neurons

To examine whether the genes in the preserved dysregulated pathways of RTT iPSCs and fibroblasts were shared in iPSC-derived RTT neurons, we compared the gene expression profile of the preserved dysregulated module of the iPSCs and iPSC-derived neurons of the OH dataset. Of the 1726 genes in the module, we identified 27 genes downregulated in the RTT iPSC-derived neurons and 13 upregulated genes (Figure 8). After performing pathway enrichment, no significant pathways were identified.

### 2.8. RTT Fibroblasts, iPSC and iPSC-Induced Neurons Share Commonly Dysregulated Genes

Of the 1824 genes in the lavenderblush preserved module identified in the of the iPSC datasets, 52 were dysregulated in the patient fibroblasts of the MT dataset, and 42 were dysregulated in iPSC-derived neurons of the OH dataset (Figure 9). Two genes, TNFAIP6 and PBK were dysregulated in all three cell lines (iPSCs, Fibroblasts and neurons) (Figure 9B). These data represent a valuable resource for determining common features of the molecular changes underpinning the pathogenesis of RTT.

## 3. Discussion

In this study, we have identified the dysregulation of protein translational and degradational pathways as well as neurotransmission changes in RTT fibroblasts, iPSCs and iPSC-derived neurons. By module construction and preservation analysis, we showed global translational defects in RTT iPSCs as well as proteasome ubiquitin dysfunction. These results are consistent with previous reports of global translational dysfunction in RTT human and murine neural cells [14] and a recent study showing decreased global translation and ribosome engagement of genes in the ubiquitination pathway, leading to the accumulation of proteins that escape proteasome degradation in RTT iPSC-derived neurons [22].

RTT is characterised by an apparently normal early development with obvious symptoms appearing between 6–18 months of age. However, subtle visible symptoms, such as hypotonia and limited social interaction, are present during early infancy, indicating that the manifestation of RTT phenotype in the brain occurs much earlier than the manifestation of the typical RTT syndromic feature. The precise mechanisms that underlie RTT pathology are still unclear; however, impaired neuronal differentiation, synaptic plasticity and neurogenesis have been implicated [3,23,24]. Studies on patient-specific stem cell-derived neurons and organoids may unveil the functional changes of the disease target cells at the different stages of disease progression. Tracking transcriptomic changes in RTT patient cells using primary cells and stem cells, opens new avenues to unveil the disease-causing mechanisms and pinpoint potential therapeutic targets for this devastating disorder.

### 3.1. Dysregulation of Protein Translational Pathways

In this study, global dysregulation of protein translation was identified in RTT iPSCs carrying a MECP2 mutation. The four predominant pathways included ribosome biogenesis in eukaryotes, endoplasmic reticulum, aminoacyl tRNA biosynthesis and the proteasome (Figure 6A), all critical pathways, essential for normal protein translation. These findings have been observed in other RTT cell types, where global transcriptional impairment has been demonstrated in post-mortem RTT human neurons [14]. Furthermore, progressive depletion of tRNA was shown to affect total cellular RNA content and was associated with downregulation of many actively transcribed mRNAs, including those for ribosomal proteins [14]. Moreover, it has also been shown that dysregulated synthetases in neurons can lead to an intracellular accumulation of misfolded proteins [25]. Nucleolar size has demonstrated to be indicative of reduced rRNA transcription observed in primary cortical neurons from a RTT mouse model [26]. More recently, global translational dysregulation related to the NEDD4 family of ubiquitin ligases in RTT iPSC-derived neurons has been reported. In these cells, a decrease in global translation and ribosomal engagement was observed, leading to the accumulation of target proteins that escape proteasome degradation [22].

### 3.2. Dysregulation of the Ubiquitin Pathway

Our study demonstrated that three of the genes (NEDL1, NEDD4L and AMRF), belonging to the highly correlated and significantly dysregulated lavenderblush module, were involved in the ubiquitination pathway. Two of these genes, namely NEDL1 and NEDD4L, which were down-regulated in RTT iPSCs, are involved in protein ubiquitin ligation and belong to the “Neuronal precursor cell-expressed developmentally downregulated 4” (NEDD4) family of E3 ubiquitin ligases, of which there are nine family members. Ubiquitination is a post-translational protein modification critical for several cellular processes and plays a crucial role in regulating proteins post-translationally. Recent reports have shown that decreased NEDD4L leads to the accumulation of target proteins that escape protein degradation in iPSC-derived neurons [26] and NEDD4L was also significantly downregulated in RTT iPSCs and fibroblasts [22,27]. The third ubiquitination related gene identified, Scavenger Receptor Class B Member 2 gene SCARB2 (AMRF), which was also downregulated, is responsible for catalysing the ubiquitination and endoplasmic reticulum-associated degradation of proteins.

Interestingly, of the 12 genes that were enriched in the proteasome pathway, 11 are related to protein deubiquitylation, suggesting that not only is there a decreased function of E3 ligases but also deubiquitinating enzymes (DUBs) [28].

### 3.3. Differentially Expressed Genes in RTT Parental Fibroblast

In our gene expression comparisons between the fibroblast gene signature and the iPSCs in the MT study (Figure 7), four ubiquitin genes (KLHL13, NEDD4L, ARRDC4 and HCN1) were observed to be downregulated in fibroblasts lines. Interestingly, NEDD4L was again shown to be dysregulated. These findings have been previously reported [27]. Many of the genes of the lavenderblush module in fibroblasts belong to inflammatory and cytoskeletal processes. This is in agreement with the literature, where MECP2 has been reported to influence the regulation of the cell cycle, cell cytoskeleton, adhesion, and extracellular matrix in different tissues [29]. In addition, chronic inflammation is widely reported in the literature, where the dysregulation of immune cells and production of pro-inflammatory signals are proposed to contribute to the development and progression of some of the clinical features of RTT (as reviewed by [30]).

### 3.4. Common Genes Identified between iPSCs, iPSC-Derived Neurons and Fibroblasts

Two genes of the lavenderblush module were observed to be dysregulated in iPSCs, iPSC-derived neurons and fibroblasts, namely TNFAIP6 (TSG-6) which as downregulated and PBK (MAPK) which was upregulated. TNFAIP6 (Tumour necrosis factor-inducible gene 6 protein) has been described to potentially be involved in cell-cell and cell-matrix interactions during inflammation and tumorigenesis whereas PBK (lymphokine-activated killer T-cell-originated protein kinase) phosphorylates MAP kinase, activates lymphoid cells and reported to be active only in mitosis [31]. To date no reports have linked MECP2 and these two genes.

TSG-6 is the secreted protein product of the TNF-stimulated gene-6 (TNFAIP6) [32] produced by mesenchymal stem cells, immune cells (e.g., neutrophils, monocytes, macrophages, myeloid dendritic cells) and by stromal cells (e.g., fibroblasts and smooth muscle cells) often in response to pro-inflammatory mediators including TNF-α and interleukin (IL)-1β [33,34,35]. TSG-6 predominantly exhibits anti-inflammatory and tissue protective properties and is produced in response to inflammatory signals [32], where it regulates the immune system, specifically leukocyte migration, macrophage polarisation, chemokine function and inflammatory signalling [35].TSG-6 released from mesenchymal stem cells ameliorates the inflammatory phenotype of microglia conferring neuroprotection and anti-neuroinflammatory effects [36]. This is particularly relevant to Rett considering inflammation has been advocated as a common central disease mechanism with MeCP2 being critical for the normal functioning of immune cells including microglia and macrophages [37]. It is unclear whether MeCP2 directly regulates the expression of TNFAIP6 or whether TNFAIP6 is dysregulated in response to the inflammatory state of neural and immune cells and further experimental evidence into how TNFAIP6 contributes to the pathology of RTT is required.

T-lymphokine-activated killer cell-originated protein kinase (TOPK, also known as PDZ-binding kinase or PBK) plays a crucial role in cell cycle regulation and mitotic progression and has mainly been described in cancer [38]. However, PBK has also been shown to be differentially expressed in normal proliferative cells, including neural precursor cells in the subventricular zone of the adult brain, as well as under pathological conditions, such as ischemic tissues, including the brain and plays important roles in their physiological functions, including proliferation and self-renewal [39]. PBK phosphorylates p38, JNK, ERK, and AKT, and activates multiple signalling pathways related to MAPK, PI3K/PTEN/AKT, and NOTCH1. MeCP2 plays an essential role during embryonic and early postnatal life when neural progenitor cells are proliferating, and early neurons are being produced. MeCP2-deficiency impairs cell fate refinement which may explain delayed cortical neuron maturation in RTT patients [40]. It is unclear whether PBK is a direct transcriptional target of MeCP2 or whether MeCP2-deficiency leads to aberrant PBK gene expression which in turn affects the normal phosphorylation of PBK targets and associated signalling pathways leading to impaired cell reprogramming of cortical neurons.

### 3.5. Limitations of Study Design and Mutation Type for Constructing Co-Expression Networks and Module Enrichment Analysis

The heterogeneity of the molecular landscape observed in this study represents an important consideration for the study of RTT in iPSC models. The three datasets analysed in this study were processed separately and identified different co-expression networks and different gene modules. The MT dataset produced seven significant modules, while the TK dataset did not generate any significant modules and the OH dataset only produced one significant module (darkorange) (Figure 4). Subsequently, the OH module was excluded from further analysis as this module was not enriched for any key biological process. The discrepancy in the number of modules being altered in the mutant cell lines between the datasets is most likely contingent on the experimental strategy of these studies: The MT study employed unaffected controls, whereas the OH and TK studies compared the patient samples to their isogenic controls. Furthermore, each study investigated the gene expression profiles of patients with different MECP2 mutations (Figure 2). The apparent heterogeneity observed between RTT and isogenic controls in the global expression profiles of TK and OH was reflected in the modules themselves and could not be reconstructed using gene co-expression modules. This recapitulates that at the IPSC stage, there are very few molecular differences between RTT and isogenic controls and that gene expression patterns can vary greatly among the different samples especially with a small sample size (e.g., [17,27,41]). Some of these differences are hypothesised to stem from the technical variability arisen in iPSC derivation and the methodology of cell differentiation [42,43]. It is important to note therefore that even with powerful gene-network based analysis, subtle molecular changes in undifferentiated cells can be eclipsed by limitations of the model and technique.

These limitations are reinforced in our study where the MT study consisting of RNA-sequencing data from two iPSC clones from the same RTT patient, with triplicates of each cell clone, was the only study that contained enough consistent differences to identify dysregulated gene modules. Nevertheless, pathway analysis and preservation analysis demonstrated that the lavenderblush module was associated with key RTT associated perturbation of protein translational pathways and ubiquitination process and was comprised of a set of conserved and correlated genes common to all three datasets. Remarkably, genes of the lavenderblush module were shown to be dysregulated in fibroblasts and iPSC-derived neural cells.

## 4. Materials and Methods

### 4.1. Data Pre-Processing

Since the data were from different profiling platforms, we performed pre-processing according to WGCNA authors instructions for the data. Briefly, deposited data for undifferentiated iPSCs were retrieved from GEO for the MT, TK and OH datasets. Raw counts and probe intensity data were pre-processed using the Limma package [44] in the R environment. Counts data were transformed by mean-variance modelling at the observational level (voom) [45] before all studies were subjected to quantile normalisation and data quality control as recommended for WGCNA. Finally, for differential gene expression [9] analysis, raw counts data were voom transformed and array data were log transformed in the Limma package [44] in the R environment, followed by quantile normalisation and data quality control.

### 4.2. Co-Expression Network Construction and Disease-Specific Module Identification

The analysis was conducted using WGCNA 1.63 source code in the R environment [19]. Detailed information on the methodology can be found in [19]. Briefly, modules were created for the three datasets independently, using the already normalised data. Network construction and module detection were analysed with the “Blockwise Modules” function in the WGCNA package. The Pearson correlation matrix was calculated for all possible RNA pairs and then transformed into an adjacency matrix with soft thresholding power using the “picksoft Threshold” function. A dynamic tree cut algorithm was used to detect groups of highly correlated genes. The minimum module size was set according to the differentially expressed gene (DEG) from each group, and the threshold for merging module was set to 0.25 as default. Thus, each module, which was assigned a unique colour, contained a unique set of genes.

### 4.3. Module-Trait Relationship

After obtaining modules from each group, module eigengene, summarised as the first principal component of expression dataset, was calculated with the “Module Eigengenes” function. The module eigengene is a weighted average of module gene expression profile. Association analysis between a module and the trait of each group was performed as the function of “corPvalueStudent” based on the module eigengene, *p*  <  0.05 was set for statistical significance. Trait was defined as disease status, understanding by this whether the samples were MECP2 mutant or wild type. The modules of interest were then selected based on having a significant correlation value (*p* < 0.05) to the trait.

### 4.4. Module Preservation Analysis

Preservation analysis was carried out for pathway enriched MT modules against OH and TK datasets using the function “modulePreservation” in the WGCNA package. This analysis utilises median rank to identify module preservation and Zsummary to assess the significance of the module via permutation testing. Based on the number of modules present in the study, a median rank of 8 was chosen as a cut-off to detect weak preservation. Permutation was performed 200 times. Based on the threshold described by Langerfeld et al., 2011 [19], modules with a Zsummary score greater than 10 indicate good preservation, 2–10 indicates weak preservation and less than 2 no preservation between datasets.

### 4.5. Module Enrichment

Associations of co-expressed genes modules and cellular pathways was performed using gene enrichment analysis in ClusterProfiler 4.0 [46]. Enrichment was performed againts the Kyoto Encyclopaedia of Genes and Genomes (KEGG) to identify enriched pathways with a Bonferroni-hochberg adjusted hypergeometric distribution *p*-value cut-off of 0.05.

### 4.6. Differential Gene Expression

Six fibroblast gene expression datasets from the MT study were analysed, which included three replicates from a patient with a 1155del32 variant and three replicates from a 24yr old wild type control (AG09319-Coriell cell line). In addition, six iPSC-derived neuron gene expression datasets from the OH study were analysed, including three (RTT mutant) and three isogenic controls derived from the same patients. Differential gene expression between the patient samples and controls was performed using the EdgeR (R Bioconductor) package [47], with a significant fold change cut-off of 1.5. Enriched genes identified in the lavenderblush module were then manually identified in each significantly dysregulated dataset.

## 5. Conclusions

RTT iPSCs, demonstrate dysregulation in the ubiquitination and proteins synthesis pathways that involve ubiquitinating and de-ubiquitinating related genes. That these genes are dysregulated in stem cells prior to neural differentiation offers new insight into the impact of MECP2 mutations. These findings are consistent with a previous report that protein degradation may be impaired not only at the ubiquitination stage but also in the de-ubiquitination process, hence leading to accumulation of proteins in iPSC-derived neurons [22]. Amongst the pleiotropic effects of MECP2 deficiency, impaired synthesis of ribosomes may be a major contributor to deficit of protein synthesis leading to cellular degeneration. A better understanding of the disease-causing mechanism, such as excessive protein accumulation, particularly in undifferentiated neurons opens the avenue for identifying amenable therapeutic targets for early treatment RTT.

## Figures and Tables

**Figure 1 ijms-22-09954-f001:**
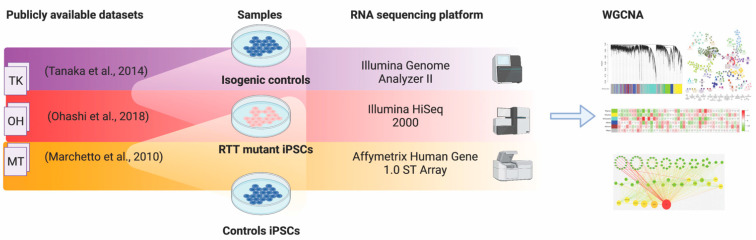
Source of RNA-sequencing datasets, sequencing platform and methodology of data analysis using Weighted Gene Correlation Network analyses (WGCNA) to identify relevant modules of genes and subsequent preservation analysis between studies.

**Figure 2 ijms-22-09954-f002:**
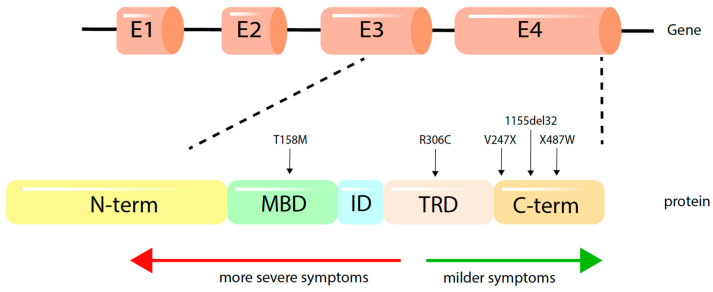
MECP2 gene and MeCP2 functional domains. Mutations covered in the present analysis are indicated by arrows. There are over 500 pathogenic mutations in MECP2, with the majority lying in exons 3 and 4 in the methyl binding domain (MBD), transcriptional repressor domain (TRD) and the C-terminal functional domain (CTD) regions, disrupting the normal function of the MeCP2 protein.

**Figure 3 ijms-22-09954-f003:**
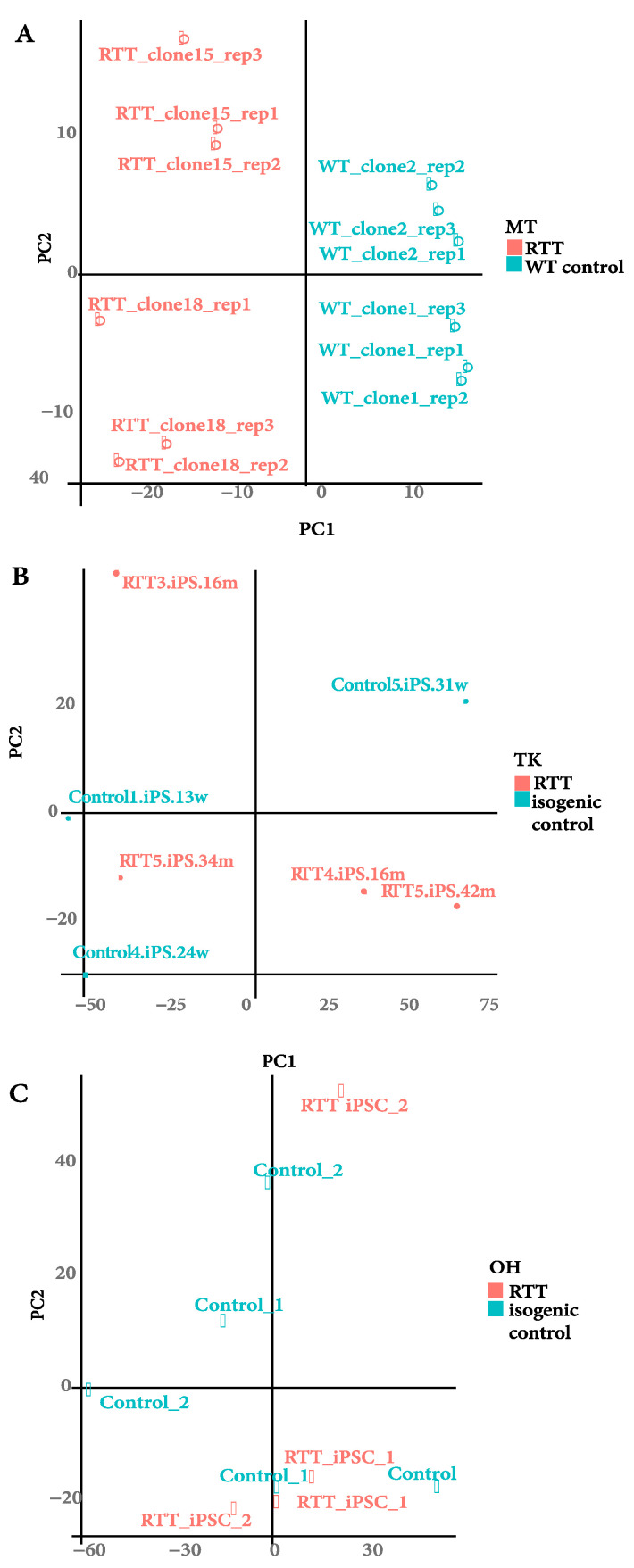
PCA plots for individual datasets showing sample distribution. (**A**). MT dataset showing sample distribution of RTT iPSCs clones 15 and 18 and wild type controls from healthy controls. (**B**). TK dataset sample distribution, RTT iPSCs and their corresponding isogenic controls. (**C**). OH dataset sample distribution, RTT iPSCs, and their isogenic controls.

**Figure 4 ijms-22-09954-f004:**
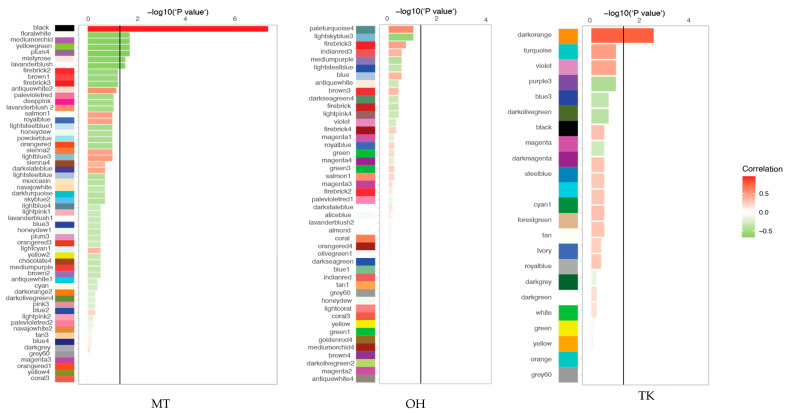
Module trait relationship plots. Colour coded gene modules identified by WGCNA for each dataset, black line depicts cut-off for significance using −log10 *p* value (*p* < 0.05). Bars are coloured in red or green according to correlation value, red being positive correlation and green negative correlation between trait and gene expression (wild type vs. mutant).

**Figure 5 ijms-22-09954-f005:**
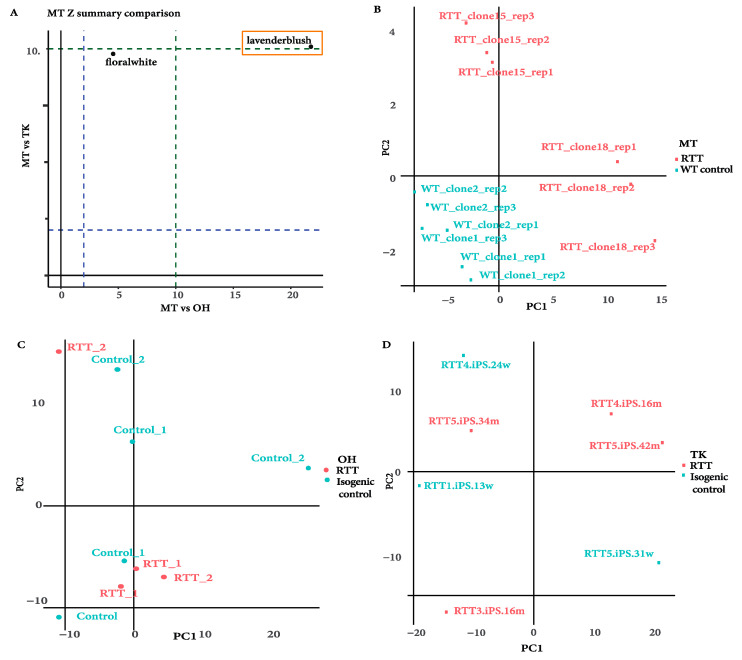
(**A**). Preservation analysis performed based on GSE21037 (MT) against GSE107399 (OH) and GSE51607 (TK). Lavenderblush represents the highest preservation score. (**B**–**D**). PCA plots for each dataset showing gene expression profiles of the lavenderblush module obtained through WGCNA, and clustering of samples according to their gene expression.

**Figure 6 ijms-22-09954-f006:**
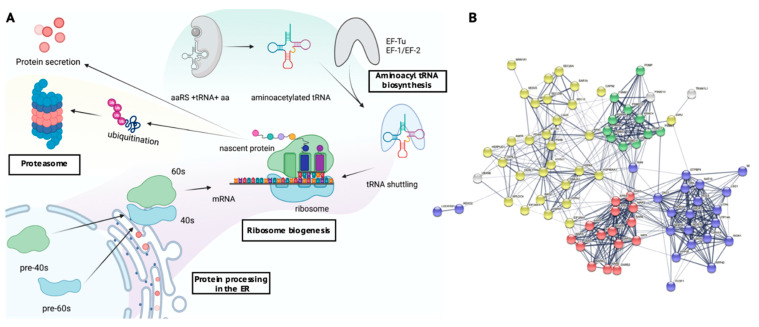
(**A**). The relationship between the four pathways in the lavenderblush module (**B**). String diagram showing gene interactions in the dysregulated pathway. String diagram depicting the four identified pathways by KEGG. Yellow shows protein processing in the ER, green: proteasome, red: aminoacyl tRNA biosynthesis and blue: ribosome biogenesis.

**Figure 7 ijms-22-09954-f007:**
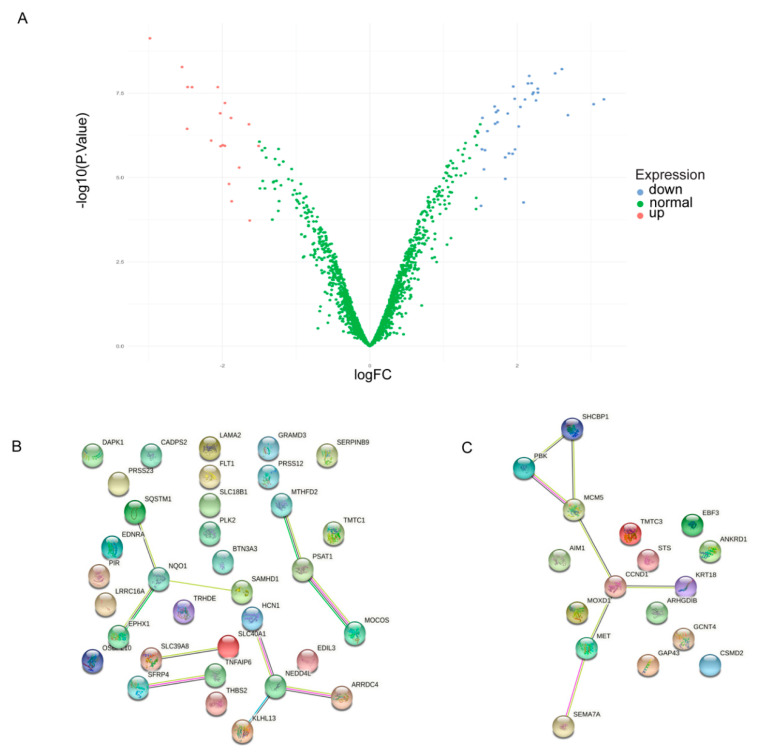
(**A**) Volcano plot showing the amount of significantly up, red, and down, blue, regulated fibroblast genes out of the lavenderblush module. String plots of down and up regulated genes from the parental fibroblasts from the MT dataset. (**B**) Downregulated genes in RTT fibroblasts samples (MT) belonging to the lavenderblush module. (**C**) Upregulated genes in RTT fibroblasts. Note: The OH and TK datasets did not include fibroblast gene expression data.

**Figure 8 ijms-22-09954-f008:**
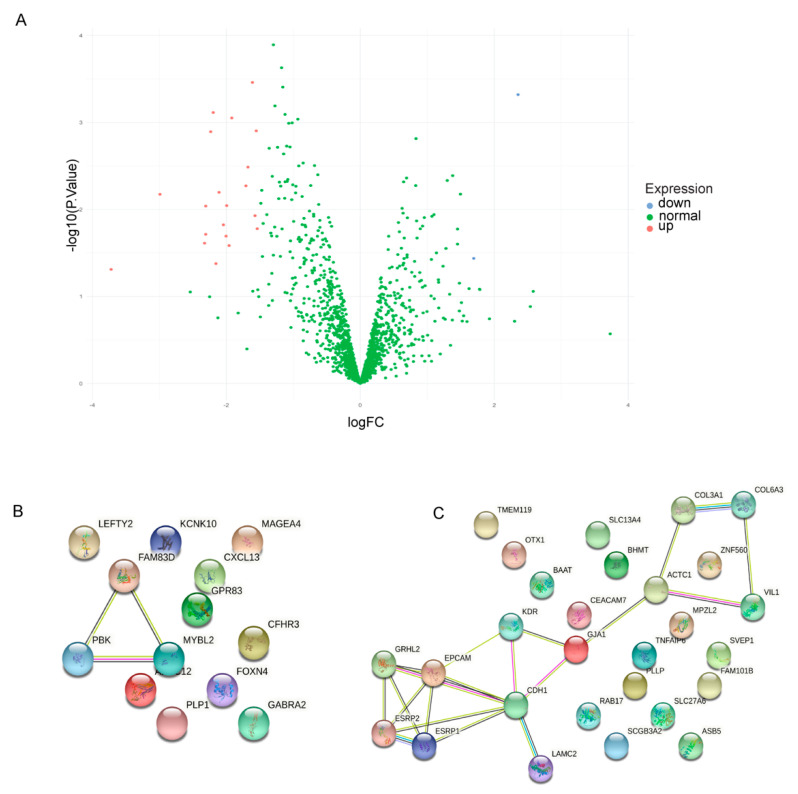
(**A**) Volcano plot showing the significantly up, red, and down, blue, regulated genes from the lavenderblush module in iPSC-derived neurons. String plots of down and up regulated genes from the iPSCs derived neurons from the OH dataset. (**B**) Upregulated genes and (**C**) Downregulated genes in RTT iPSCs-derived neurons. Note: The MT and TK datasets did not include iPSC-differentiated neuronal gene expression data.

**Figure 9 ijms-22-09954-f009:**
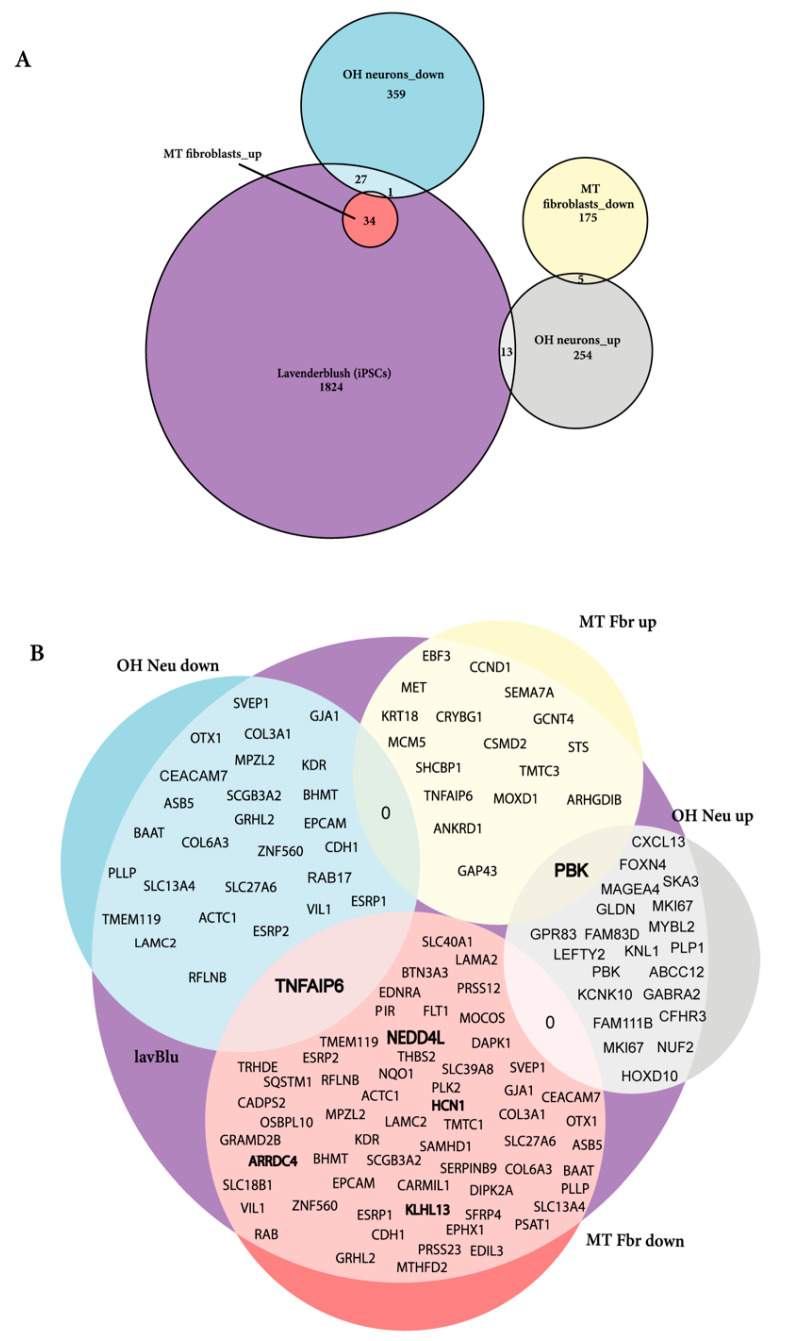
Venn diagram demonstrating the differentially expressed genes in iPSC-derived neurons and fibroblasts. (**A**) The number of genes in each data set and whether they are up or down regulated are shown where the size of the circles represents the relative proportion of those genes that correspond to the genes in the preserved module. (**B**) Diagram showing the genes preserved from lavendersblush across neurons and fibroblasts. Fbr up/down stand for fibroblast (MT) genes up or down regulated (bolded genes represent ubiquitination related genes), OH Neu down/up represent the iPSCs derived neurons (OH) and lavBlu represents the lavenderblush module obtained in the iPSCs.

**Table 1 ijms-22-09954-t001:** Summary table of studies used in the present study, including sample ID, status and mutation (protein change and nucleotide change).

	Sample ID	Status	Mutation (Protein Change)	Mutation (Nucleotide Change)
TK GSE51607 Tanaka et al.	Control1.iPS.13w	Isogenic	p.T158M	c. 473C>T
	Control5.iPS.31w	Isogenic	p.X487W	c.1461A>G
	Control4.iPS.24w	Isogenic	p.R306C	c.916C>T
	RTT5.iPS.42m	Disease	p.X487W	c.1461A>G
	RTT5.iPS.34m	Disease	p.X487W	c.1461A>G
	RTT3.iPS.16m	Disease	E235fs	705delG
	RTT4.iPS.16m	Disease	p.R306C	c.916C>T
OH GSE107399 Ohashi et al.	Control	WT	-	
	Control 1 (rep 1 and 2)	Isogenic	p.V247X	c.705delG
	Control 2 (rep 1 and 2)	Isogenic	p.V247X	c.705delG
	RTT_1 (rep1 and 2)	Disease	p.X487W	c.1461A>G
	RTT_2 (rep1 and 2)	Disease	p.V247X	c.705delG
MT GSE21037 Marchetto et al.	Cnl15Rep1	Disease	p.L386Rfs	c.1155del32
	Cln15Rep2	Disease	p.L386Rfs	c.1155del32
	Cln15Rep3	Disease	p.L386Rfs	c.1155del32
	Cln18Rep1	Disease	p.L386Rfs	c.1155del32
	Cln18Rep2	Disease	p.L386Rfs	c.1155del32
	Cln18Rep3	Disease	p.L386Rfs	c.1155del32
	WT Cln1Rep1	Paedriatic control	-	-
	WT Cln1Rep1	Paedriatic control	-	-
	WT Cln1Rep3	Paedriatic control	-	-
	WT Cln2Rep1	Paedriatic control	-	-
	WT Cln2Rep2	Paedriatic control	-	-
	WT Cln2Rep3	Paedriatic control	-	-

**Table 2 ijms-22-09954-t002:** Pathways with KEGG enrichment identified in lavenderblush.

ID	Description	Gene Ratio	*p* Value	*p* Adjust	*q* Value	Gene ID	Count
**hsa03008**	Ribosome biogenesis in eukaryotes	22/729	3.00 × 10^−6^	0.00094864	0.00092589	REXO5/NOL6/LSG1/NOP58/GTPBP4/UTP6/WDR75/RIOK1/WDR43/WDR3/REXO2/NOP56/MDN1/GNL3L/DKC1/RAN/UTP4/RPP30/UTP14A/NAT10/TCOF1/RPP40	22
**hsa00970**	Aminoacyl-tRNA biosynthesis	15/729	1.48 × 10^−5^	0.00233317	0.00227721	KARS/RARS/LARS/EPRS/HARS/IARS/GARS/TARS/DARS2/NARS/YARS/WARS/AARS/MARS/NARS2	15
**hsa04141**	Protein processing in endoplasmic reticulum	32/729	0.00036708	0.03107943	0.03033403	AMFR/HSPA2/EIF2AK4/HSP90AA1/SEC24A/HSPA5/HERPUD1/DNAJA1/SEC23B/UBXN8/HSPA8/BAG2/MOGS/HSPH1/DERL1/SEC24D/CAPN2/SSR2/RPN1/LMAN2/EIF2AK3/TRAM1L1/CALR/DERL2/SAR1A/MAN1A1/HYOU1/SEC13/NPLOC4/STT3A/PDIA4/P4HB	32
**hsa03050**	Proteasome	12/729	0.00039341	0.03107943	0.03033403	POMP/PSMD13/PSMC5/PSMA4/PSMD7/PSMD11/PSMD12/PSMB1/PSMC4/PSMA7/PSME3/ PSMA5	12

**Table 3 ijms-22-09954-t003:** Descriptions of the four genes related to ubiquitination (KLHL13, NEDD4L, ARRDC4 and HCN1) that are downregulated in RTT fibroblasts (information from UniProt).

Gene	Function
**KLHL13**	Kelch-like protein 13; Substrate-specific adapter of a BCR (BTB-CUL3-RBX1) E3 ubiquitin-protein ligase complex required for mitotic progression and cytokinesis. The BCR(KLHL9-KLHL13) E3 ubiquitin ligase complex mediates the ubiquitination of AURKB and controls the dynamic behaviour of AURKB on mitotic chromosomes and thereby coordinates faithful mitotic progression and completion of cytokinesis.
**NEDD4L**	E3 ubiquitin-protein ligase NEDD4-like; E3 ubiquitin-protein ligase which accepts ubiquitin from an E2 ubiquitin-conjugating enzyme in the form of a thioester and then directly transfers the ubiquitin to targeted substrates. Inhibits TGF-beta signalling by triggering SMAD2 and TGFBR1 ubiquitination and proteasome-dependent degradation. Promotes ubiquitination and internalization of various plasma membrane channels such as ENaC, Nav1.2, Nav1.3, Nav1.5, Nav1.7, Nav1.8, Kv1.3, KCNH2, EAAT1 or CLC5.
**ARRDC4**	Arrestin domain-containing protein 4; Functions as an adapter recruiting ubiquitin-protein ligases to their specific substrates (By similarity). Plays a role in endocytosis of activated G protein-coupled receptors (GPCRs) (Probable). Through an ubiquitination-dependent mechanism plays also a role in the incorporation of SLC11A2 into extracellular vesicles (By similarity). May play a role in glucose uptake.
**HCN1**	Potassium/sodium hyperpolarization-activated cyclic nucleotide-gated channel 1; Hyperpolarization-activated ion channel exhibiting weak selectivity for potassium over sodium ions. Contributes to the native pacemaker currents in heart (If) and in neurons (Ih).

## Data Availability

Code provided upon request.

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
