# Peer review of "WGCNA Identifies Translational and Proteasome-Ubiquitin Dysfunction in Rett Syndrome"

_ijms, 2021, doi:10.3390/ijms22189954_

Round 1

Reviewer 1 Report

Thank you for allowing me to review the manuscript by Haase, and colleagues “WGCNA identifies translational and proteasome-ubiquitin dysfunction in Rett syndrome” an interesting study describing the use of weighted gene correlation Network Analysis to study transcriptomic changes in Rett syndrome iPSCs RNA-sequencing dataset.  

Major revisions

The databases used for the analysis should be more clearly explained. In the introduction the authors mentioned MT (line 80) but they did not mention the referring dataset. I suggest to pinpoint which mutation are covered by different databases.

How many genes mapped into different modules of MT dataset? Lavenderblush consists of 1,824 genes. The preservations score was high in all datasets. Might this happen because the number of genes-related lavenderblush network are low compared to other modules?  

Might the authors better explain the preservation analysis? It’s not clear to me how a high preservation score could be associated to lack of stratification at PCA.

The results regarding MT dataset is just valid for one mutation of one Rett patient. I think the results should not be generalized as presymptomatic of Rett syndrome patients and toned down the results section.

Please clarify “their expression changes in parental and iPSC-derived expression profile” line 207 and 208.  

It’s not clear to me why NEDD4L perturbation was a clue for pre-symptomatic and symptomatic cells.

Which settings have been used in String (interaction score..)?

Minor revisions

Properly use the brackets see lines 76, and 81.

The labels of figure 5 were not of the same size.

In the text Table 1 is cited after Table 2. I do not see Table 2, while two Table 1. Please correct.

Some references are showed as [25] [26] some other [3,23,24].

MECP2 when referring a gene should be written in italics (line 286).

Line 288: This findings --> these findings

Author Response

Thank you very much for reviewing our study, we have attached the responses below. 

Reviewer 2 Report

The authors performed an analysis of publicly available RNA-sequencing datasets of iPSCs, iPSC-derived neurons, and iPSC-parental cells (fibroblasts) with MECP2 pathogenic mutations from patients with Rett syndrome and healthy controls, by applying several methods including Weighted Gene Correlation Analysis (WGCNA).  The focus of the analysis was to identify pre-symptomatic transcriptomic changes.  It appears that new molecular mechanisms reported in recent publications, which implicate protein synthesis, ribosomal function and ubiquitination in Rett syndrome, were also a secondary focus of the study.

WGCNA followed by preservation analysis and differential gene expression identified gene pathways involved in translation, ribosomal function, and ubiquitination perturbed in iPSCs, and abnormal expression in ubiquitination pathway genes and genes involved in neurotransmission in fibroblasts and iPSC-derived neurons, respectively.

The study presents multiple strengths including optimization of acquired data use, innovative analytical strategy for complex co-expression data, and comparisons between the three ‘sequential’ cell types involved in neural differentiation.  Illustrations are also excellent quality and complement the text well.

However, there multiple weaknesses.  The first is the conceptual framework.  Molecular events preceding differentiation to neural cell lineage may reflect pre-symptomatic abnormalities but may not.  Neural differentiation could begin before clinical manifestations.  In humans, neuronal cortical differentiation begins in the second trimester and extends into the 4th-6th year of life because of its complex temporospatial patterns.  And this does not take into consideration other early neural differentiation processes.  The last paragraph of the Introduction is cautious in this regard, but this caution is not present throughout the manuscript.  The Discussion mentions progressive neuronal differentiation, which is not a feature of Rett syndrome.  Although other processes can also be affected (e.g., neurogenesis), Rett syndrome is primarily a disorder of neuronal differentiation.

Other weaknesses are related to data presentation and interpretation. First, the common features of the three cell types (i.e. two dysregulated genes of the lavenderblush module) are underplayed, not even mentioned in the Conclusions, apparently because of the emphasis on ‘pre-symptomatic’ events.  Also, the conclusion at the end of the Introduction that ‘These findings point to global dysregulation of translation…’ implies that proteasome-ubiquitin function is a component of protein synthesis, which is not.  Finally, the search for the genetic basis of other neurodevelopmental disorders, in particular autism, has revealed that not only genes involved directly or indirectly in synaptic function play a critical role but the proteasome system is also a key component.  The discussion of the study findings is too Rett-centric, too focused on recent literature on protein synthesis dysregulation in the disorder.  The Discussion should cover relevant findings in other neurodevelopmental disorders, including the mTOR pathway and protein synthesis, among others, in fragile X syndrome and tuberous sclerosis.

Minor issues include the lack of explanation for the use of MT, OH, and TK as labels for the datasets.

Author Response

Reviewer 2

Comments to authors

The authors performed an analysis of publicly available RNA-sequencing datasets of iPSCs, iPSC-derived neurons, and iPSC-parental cells (fibroblasts) with MECP2 pathogenic mutations from patients with Rett syndrome and healthy controls, by applying several methods including Weighted Gene Correlation Analysis (WGCNA).  The focus of the analysis was to identify pre-symptomatic transcriptomic changes.  It appears that new molecular mechanisms reported in recent publications, which implicate protein synthesis, ribosomal function, and ubiquitination in Rett syndrome, were also a secondary focus of the study.

WGCNA followed by preservation analysis and differential gene expression identified gene pathways involved in translation, ribosomal function, and ubiquitination perturbed in iPSCs, and abnormal expression in ubiquitination pathway genes and genes involved in neurotransmission in fibroblasts and iPSC-derived neurons, respectively.

The study presents multiple strengths including optimization of acquired data use, innovative analytical strategy for complex co-expression data, and comparisons between the three ‘sequential’ cell types involved in neural differentiation.  Illustrations are also excellent quality and complement the text well.

However, there multiple weaknesses.  The first is the conceptual framework.  Molecular events preceding differentiation to neural cell lineage may reflect pre-symptomatic abnormalities but may not.  Neural differentiation could begin before clinical manifestations.  In humans, neuronal cortical differentiation begins in the second trimester and extends into the 4th-6th year of life because of its complex temporospatial patterns.  And this does not take into consideration other early neural differentiation processes.  The last paragraph of the Introduction is cautious in this regard, but this caution is not present throughout the manuscript.  The Discussion mentions progressive neuronal differentiation, which is not a feature of Rett syndrome.  Although other processes can also be affected (e.g., neurogenesis), Rett syndrome is primarily a disorder of neuronal differentiation.

Thank you very much for your feedback. We agree with the reviewers and have carefully revised the appropriate sections to to indicate the transcriptomic changes we observed are due to undifferentiated cells and not to pre-symptomatic cells.

We agree with the reviewers’ comments regarding progressive neuronal degeneration and have changed the text. This was na auto-correct mistake in the final editing and we meant to use differentiation and not degeneration.

“The precise mechanisms that underlie RTT pathology are still unclear; however, impaired neuronal differentiation, synaptic plasticity and neurogenesis have been implicated.”

Other weaknesses are related to data presentation and interpretation. First, the common features of the three cell types (i.e. two dysregulated genes of the lavenderblush module) are underplayed, not even mentioned in the Conclusions, apparently because of the emphasis on ‘pre-symptomatic’ events:

We agree with the reviewer that the role of the two common genes found has been underplayed in the manuscript, hence we have added more information in the discussion section:

“TSG-6 is the secreted protein product of the TNF-stimulated gene-6 (TNFAIP6) [33] produced by mesenchymal stem cells, immune cells (e.g., neutrophils, monocytes, macrophages, myeloid dendritic cells) and by stromal cells (e.g., fibroblasts and smooth muscle cells) often in response to pro-inflammatory mediators including TNF-α and in-terleukin (IL)-1β [34,35] [36]. TSG-6 predominantly exhibits anti-inflammatory and tissue protective properties and is produced in response to inflammatory signals [33], where it regulates the immune system, specifically leukocyte migration, macrophage polarisation, chemokine function and inflammatory signalling [36].TSG-6 released from mesenchymal stem cells ameliorates the inflammatory phenotype of microglia conferring neuroprotec-tion and anti-neuroinflammatory effects [37]. This is particularly relevant to Rett con-sidering inflammation has been advocated as a common central disease mechanism with MeCP2 being critical for the normal functioning of immune cells including microglia and macrophages [38]. It is unclear whether MeCP2 directly regulates the expression of TNFAIP6 or whether TNFAIP6 is dysregulated in response to the inflammatory state of neural and immune cells and further experimental evidence into how TNFAIP6 con-tributes to the pathology of RTT is required.

T-lymphokine-activated killer cell-originated protein kinase (TOPK, also known as PDZ-binding kinase or PBK) plays a crucial role in cell cycle regulation and mitotic progression and has mainly been described in cancer [39]. However, PBK has also been shown to be differentially expressed in normal proliferative cells, including neural precursor cells in the subventricular zone of the adult brain, as well as under pathological conditions, such as ischemic tissues, including the brain and plays important roles in their physiological functions, including proliferation and self-renewal [40]. PBK phosphorylates p38, JNK, ERK, and AKT, and activates multiple signalling pathways related to MAPK, PI3K/PTEN/AKT, and NOTCH1. MeCP2 plays an essential role during embryonic and early postnatal life when neural progenitor cells are proliferating, and early neurons are being produced. MeCP2-deficiency impairs cell fate refinement which may explain delayed cortical neuron maturation in RTT patients [41]. It is unclear whether PBK is a direct transcriptional target of MeCP2 or whether MeCP2-deficiency leads to aberrant PBK gene expression which in turn affects the normal phosphorylation of PBK targets and associated signalling pathways leading to impaired cell reprogramming of cortical neurons.”

Also, the conclusion at the end of the Introduction that ‘These findings point to global dysregulation of translation…’ implies that proteasome-ubiquitin function is a component of protein synthesis, which is not. 

We have clarified the statement ‘These findings point to global dysregulation of translation…’, to: ‘These findings suggest that global translational dysregulation and proteasome ubiquitin function in Rett syndrome…’

Finally, the search for the genetic basis of other neurodevelopmental disorders, in particular autism, has revealed that not only genes involved directly or indirectly in synaptic function play a critical role but the proteasome system is also a key component.  The discussion of the study findings is too Rett-centric, too focused on recent literature on protein synthesis dysregulation in the disorder.  The Discussion should cover relevant findings in other neurodevelopmental disorders, including the mTOR pathway and protein synthesis, among others, in fragile X syndrome and tuberous sclerosis.

We acknowledge the comment on the inclusion of other neurodevelopmental disorders, however the focus of this study is on Rett syndrome cells harbouring mutations in the MECP2 gene and we feel this is too premature to extrapolate these findings to other neurodevelopmental disorders. Furthermore, as this manuscript in being submitted as part of a special edition on Rett syndrome we believe it is important to be Rett-centric

Minor issues include the lack of explanation for the use of MT, OH, and TK as labels for the datasets.

We have included a table in the text (Table 1) as suggested by reviewer 1 and clarification across the figures.

Comments to editor

This is an interesting and contributory study.  However, it needs more refinement before it is ready for publication.  Also, goals and conclusions should be tempered and placed in a more global neurodevelopmental perspective.

Reviewer 3 Report

The study identifies translational and proteasome-ubiquitin dysfunction in Rett syndrome using a transcriptomics approach, analysing patient-derived iPSCs. The manuscript is well written and adds to the body of evidence in the literature that translation, ribosomal function and ubiquitination are affected in RTT patients.

A major conclusion of the study is the identification of pre-symptomatic transcriptomic changes in RTT patients. Can the authors elaborate on this more? Given the overlap in conclusions with the 2020 paper by Rodrigues et al. – “Shifts in Ribosome Engagement Impact Key Gene Sets in Neurodevelopment and Ubiquitination in Rett Syndrome” – are the iPSC datasets analysed in the submitted manuscript more representative of pre-symptomatic RTT cells than iPSC datasets analysed in other published studies?

In the conclusions, the authors state that “A better understanding of the disease-causing mechanism, such as excessive protein accumulation, particularly at the pre-symptomatic stage opens the avenue for identifying amenable therapeutic targets for early treatment RTT.” Given that almost all MECP2 mutations occur de novo, and therefore molecular diagnosis always occurs post-symptomatically, can the authors provide a qualification to this statement.

Additional comments:

To help the reader, it would be helpful to include a brief table in the main text describing the datasets. Although there is a comprehensive table in supplementary, it would be extremely helpful to the reader to have a reference table in the main text – to state unequivocally what the MT, TK and OH groups mean with respect to the datasets, and what samples are analysed in those groups.

The OH dataset in Figures 3 and 5 do not match the nomenclature in Supplementary Table 1 – this should be corrected so it is clear which samples are which in the figures.

In the TK dataset in Figures 3 and 5 there is a sample – RTT5.iPS.34m – that does not appear to be listed in Supplementary Table 1.

Figure 5 – the axis labels should be made consistent for font and font size – in B and D the axis titles and scales are currently too small to read clearly.

The text labels in the volcano plots (Figures 7A & 8A) are also too small and should be increased to be more legible.

Figure 7 – there appears to be a typo in the figure legend – should be ‘blue’ instead of a ref ‘13’.

In Supplementary Table 1, MECP2 mutations are listed as a mixture of protein changes and nucleotide changes – it would be helpful if there were two columns – one for each, listing the nucleotide change and the protein change.

Author Response

Reviewer 3

The study identifies translational and proteasome-ubiquitin dysfunction in Rett syndrome using a transcriptomics approach, analysing patient-derived iPSCs. The manuscript is well written and adds to the body of evidence in the literature that translation, ribosomal function and ubiquitination are affected in RTT patients.

A major conclusion of the study is the identification of pre-symptomatic transcriptomic changes in RTT patients. Can the authors elaborate on this more? Given the overlap in conclusions with the 2020 paper by Rodrigues et al. – “Shifts in Ribosome Engagement Impact Key Gene Sets in Neurodevelopment and Ubiquitination in Rett Syndrome” – are the iPSC datasets analysed in the submitted manuscript more representative of pre-symptomatic RTT cells than iPSC datasets analysed in other published studies?

Thank you very much for your comments.

We are unable to determine whether the iPSCs analysed in our datasets are more representative of presymptomatic cells than to the iPSC datasets analysed by other as currently the only available transcriptomic data of pre-symptomatic RTT cells is from non-human studies as RTT is predominantly caused by de novo mutations and therefore the pre-symptomatic stage in humans is mostly never identified. We agree with the reviewers that some parts of the manuscript need some more caution with the use of iPSCs as a model for pre-symptomatic cells, hence we have made corrections through the text to keep this consistent and to demonstrate that our results reflect undifferentiated cells rather than presymptomatic cells.

In terms of the iPSCs datasets analysed in the present study and the ones used by Rodrigues et al., we had the following mutations: pT158M, p.X487W, p.R306C, p.E235fs (TK) c.1461A>G, c.705delG (OH)  and c.1155del32(MT). The module construction was based on MT and they used a truncating mutation. Rodrigues et al., used the following mutation g.61340_67032delinsAGTTGTGCCAC and g.67072_67200del, p.T158M, p.R306C, RTTe1 - NM_001110792.1/c.47_57del, p.G16Efs*22. In their study Rodrigues et al showed decreased levels of NEDD4L protein in iPSCs derived neurons however not in the undifferentiated iPSCs. Here we show that NEDD4L is downregulated already at the iPSC level. This difference can be attributed to experimental design, patient mutation used as well as the different methods used to measure NEDD4L gene expression and protein expression.

In the conclusions, the authors state that “A better understanding of the disease-causing mechanism, such as excessive protein accumulation, particularly at the pre-symptomatic stage opens the avenue for identifying amenable therapeutic targets for early treatment RTT.” Given that almost all MECP2 mutations occur de novo, and therefore molecular diagnosis always occurs post-symptomatically, can the authors provide a qualification to this statement.

The reviewer is correct that most mutations are de novo and therefore molecular diagnosis occurs post-symptomatically.   An understanding of the early stages of disease opens many avenues for therapeutic targets for individuals with RTT. Firstly, although not available yet, pre-natal testing is on the horizon for many genetic neurological disorders, including MECP2-mutations and once this is implemented early intervention will be critical. Secondly, Rett is a progressive disorder, and we are proposing that with a better understanding of the early changes and cellular dysfunction that occurs in RTT, the progression may be halted, slowed down or even reversed before irreversible damage is caused during the critical early stages of post-natal development. Despite our opinion on this we have changed the language we have used in this study for the transcriptomic analysis to reflect undifferentiated cells and not pre-symptomatic cells.

Additional comments:

To help the reader, it would be helpful to include a brief table in the main text describing the datasets. Although there is a comprehensive table in supplementary, it would be extremely helpful to the reader to have a reference table in the main text – to state unequivocally what the MT, TK and OH groups mean with respect to the datasets, and what samples are analysed in those groups.

Thank you for this suggestion.

We have included a summary table in the text (Table 1) to complement the supplementary table.

The OH dataset in Figures 3 and 5 do not match the nomenclature in Supplementary Table 1 – this should be corrected so it is clear which samples are which in the figures.

We have now corrected the nomenclature to keep the sample names consistent.

In the TK dataset in Figures 3 and 5 there is a sample – RTT5.iPS.34m – that does not appear to be listed in Supplementary Table 1.

The sample name RTT5.iPS.34m has been added to both tables

Figure 5 – the axis labels should be made consistent for font and font size – in B and D the axis titles and scales are currently too small to read clearly.

Thank you, we have corrected this now.

The text labels in the volcano plots (Figures 7A & 8A) are also too small and should be increased to be more legible.

Thank you, we have corrected this now.

Figure 7 – there appears to be a typo in the figure legend – should be ‘blue’ instead of a ref ‘13’. This has been corrected in the legend.

In Supplementary Table 1, MECP2 mutations are listed as a mixture of protein changes and nucleotide changes – it would be helpful if there were two columns – one for each, listing the nucleotide change and the protein change.

Thank you for the feedback, we agree this will be more helpful for the reader and we have now included both forms (p. and c.) in the mutation columns.
